# Automated dry thawing of cryopreserved haematopoietic cells is not adversely influenced by cryostorage time, patient age or gender

Peter Kilbride[1]⊙*, Julie Meneghel[1]⊙*, Giovanna Creasey[1], Fatemeh Masoudzadeh[1], Tina Drew[2], Hannah Creasey[3], David Bloxham[3], G. John Morris[1], Kevin Jestice[2]

1 Cytiva, Danaher Corporation, Cambridge, United Kingdom, 2 Cambridge Cellular Therapy Laboratory, Clinical Haematology, Cambridge University Hospitals NHS Foundation Trust, Cambridge, United Kingdom, 3 Haematopathology and Oncology Diagnostic Service, Cambridge University Hospitals NHS Foundation Trust, Cambridge, United Kingdom

⊙ These authors contributed equally to this work.
* peter.kilbride@cytiva.com (PK); julie.meneghel@cytiva.com (JM)

**Data Availability Statement:** All relevant data are available within a repository at https://synapse.org/apheresisthawingdataset.

## Abstract

Cell therapies are becoming increasingly widely used, and their production and cryopreservation should take place under tightly controlled GMP conditions, with minimal batch-to-batch variation. One potential source of variation is in the thawing of cryopreserved samples, typically carried out in water baths. This study looks at an alternative, dry thawing, to minimise variability in the thawing of a cryopreserved cell therapy, and compares the cellular outcome on thaw. Factors such as storage time, patient age, and gender are considered in terms of cryopreservation and thawing outcomes. Cryopreserved leukapheresis samples from 41 donors, frozen by the same protocol and stored for up to 17 years, have been thawed using automated, water-free equipment and by conventional wet thawing using a water bath. Post-thaw viability, assessed by both trypan blue and flow cytometry, showed no significant differences between the techniques. Similarly, there was no negative effect of the duration of frozen storage, donor age at sample collection or donor gender on post-thaw viability using either thawing method. The implication of these results is that the cryopreservation protocol chosen initially remains robust and appropriate for use with a wide range of donors. The positive response of the samples to water-free thawing offers potential benefits for clinical situations by removing the subjective element inherent in water bath thawing and eliminating possible contamination issues.

## Introduction

The use of cell therapies such as CAR T cells as an effective treatment for a range of conditions is growing rapidly, harnessing the power of the immune system to fight cancers [1]. Sourcing the initial biological sample to create the preparation used for treatment is the first, key element in this process. For blood-based therapies this is commonly taken from cord blood, or an apheresis sample, for autologous treatments or allografts [2]. The initial sample may be

**Funding:** The funder Cytiva provided support in the form of salaries only for authors PK, GC, JM, and GJM, but did not have any additional role in the study design, data collection and analysis, decision to publish, or preparation of the manuscript. The specific roles of these authors are articulated in the 'author contributions' section.

**Competing interests:** Authors PK, JM, GC, and GJM are employees of Cytiva, which provided salaries for these authors. No consultancy, patents, products in development, or marketed products were derived from this study. This does not alter our adherence to PLOS ONE policies on sharing data and materials.

minimally manipulated e.g. by apheresis or may become the starting point of a more complex manufacturing process to provide the final therapeutic material, a common feature of CAR T treatments [3].

The processes for transforming an initial sample into completed cell materials, and inevitable uncertainties over the time and place of delivery to the patient, makes effective storage an essential, enabling element in effective treatment [4]. Cryopreservation offers stable, extended storage and samples can be cryopreserved immediately after extraction e.g. for cord blood and then stored in a cell bank until required [5]. Additional processing of an initial sample can also take place before cryopreservation e.g. leukapheresis of sample from patients in remission from myeloma or non-Hodgkin's lymphoma, for use if the patient relapses.

Cryopreservation has three key phases, notably cooling, storage and thawing [4,6]. For clinical cell systems, beyond the research laboratory, the first two of these are precisely controlled and recorded using validated protocols and automated controlled-rate freezers. Automatic alarms and monitoring systems are essential for good storage practice in frozen tissue banks [7]. Efficient thawing, with minimal reduction in viability and performance is essential before further processing and is often the final manipulation of a completed product carried out at the point of delivery to the patient. Clearly, any errors in thawing that damage the product, however they are caused, could have damaging consequences for the effectiveness of the cell therapy.

Thawing of cryopreserved materials has developed, over time, as a relatively simple procedure with a strong, subjective element. Typically, this involves the immersion of the frozen sample in a water bath at 37°C with melting of the last ice visually determined (wet thawing). Different operators may choose a slightly different end-of-thaw indicator, with samples e.g. cryobags, held at different angles or agitated at different speeds (or not at all). This compounds the risk of user-to-user variability producing variable results. It is acknowledged that, in the hands of a specialist technician, the essentially subjective technique of wet thawing is successful and, largely, consistent. However, the end-user of a cryopreserved product can be separated by location and time from the specialists that processed and froze the initial material. Consequently, thawing is increasingly carried out, often at the bedside, by clinical staff who may have little, or no, training or experience in cryopreservation. This generates a real risk of mishandling that can reduce post-transplant performance, due to a reduction in viable cell number.

Whilst practicable in a research laboratory, thawing water baths can also create a contamination risk that is unacceptable in many clinical situations [8–10]. Additional time and facilities for sterilisation, rewarming, refilling and temperature stabilisation must also be available. Recently, however, variants of equipment that enable water-free thawing of larger samples, held in cryobags, are becoming available. These systems use mechanical heating, such as a warm metallic plate as used in this study and/or warmed but sealed liquids that do not come into direct contact with the sample being thawed [11–15]. These systems eliminate user-to-user variability and provide a consistent, programmable process that removes any subjective intervention on the part of the user. They also provide options for computerised control, monitoring and data recording. Previous studies have also indicated that dry thawing can be applied successfully to non-cellular therapeutic materials such as plasma samples [11–13,15], however as water is fluid and a very effective thermal conductor, these typically have slightly longer warming times than a water bath-based system.

Patients selected for apheresis, including leukapheresis, for myeloma therapy, will show innate, individual variation in responses to mobilization possibly due to age, health condition or gender [16,17]. To provide therapy, using cryopreserved material, at an optimal level it is essential to understand how this variation may influence the post-thaw performance of thawed cell preparations. Any further increase in variation that could be caused by poor

control within the cryopreservation process has to be minimised. This is particularly relevant to thawing for the reasons outlined above.

The availability of cryopreserved leukapheresis samples destined for disposal (taken from donors who had been successfully treated for myeloma and were in remission), provided a unique opportunity to compare and review the post-thaw performance of samples that had been stored, using the same protocol, for as long as 17 years. The protocol used was able to cryopreserve the $2x10^6$ viable cells $kg^{-1}$ (measured pre-cryopreservation) of recipient body weight in most patients deemed necessary for effective therapy [16,17]. The study used paired samples for up to 41 patients and the influence on post-thaw viability of cryostorage time, patient gender and age at sample collection was investigated. Additionally, the study compared the effectiveness of water-free and wet thawing on these samples.

## Materials and methods

### Cell samples and cryopreservation

Paired leukapheresis samples from male and female patients in remission from myeloma or non-Hodgkin's lymphoma, aged between 39 and 70 years old at collection were provided. The mobilized peripheral blood was prepared by mobilization techniques, which include five daily injections of filgrastim (G-CSF) and cyclophosphamide to stimulate stem cells out of bone marrow into the bloodstream.

Apheresis samples were obtained post-discard from the biobank which were no longer needed for clinical use. Patients previously gave informed consent for cell donations to the cell bank to be used for research and development if they were no longer required for clinical treatment.

There were between 60 and 140ml of the completed cell preparation in each cryobag (CryoStore, CS500NS or CS250NS, Origen Biomedical, Austin, USA). The samples were double bagged with an overwrap (Seaborn Laminate Polypropylene Pouch, Moore & Buckle, St Helens, UK).

The bags had been cooled in a Kryo-10 Planer controlled-rate freezer (Planer, Sudbury, UK) following a protocol using a classical set of cooling rates for the cryopreservation of haematopoietic stem cells: a 10-minute equilibration at 4°C in cryoprotectant consisting of 10% DMSO in 4.5% Human Albumin Serum, followed by a 2°C $min^{-1}$ cooling rate down to -30°C, raised to 4°C $min^{-1}$ [18–21]. Samples were cooled to -100°C before transfer to the vapour phase above liquid nitrogen for storage. The cooling profile was recorded for each cryopreservation run. Continuous temperature monitoring was in place to ensure that the samples did not experience any warming during storage. All sample pairs were cryopreserved from the same apheresis, during the same cryopreservation run with equal volumes per bag.

### Thawing

Prior to thawing, cryobags were directly transferred from the storage vessel into a fully charged dry shipper (Chart MVE, Ball Ground, GA, USA) to facilitate transfer to the thawing laboratory. Continuous temperature monitoring was employed during transfer to ensure the integrity of the cryochain. A pair of bags from the same patient extraction were thawed concurrently, one in a standard laboratory water bath (wet thawing), and the other in a water-free system. The post-thaw tests on each pair of bags were also carried out concurrently.

To wet thaw, a 16-litre non-circulating water bath with thermostatic temperature control was freshly filled with water less than 1h before each event and was monitored as being within 1°C of 37°C before use. The temperature was monitored with type T thermocouples connected to a TC-08 temperature measuring unit (Picotechnology, St. Neots, UK). A cryobag was removed

from the dry shipper and immediately fully submerged in the water bath, where it was gently agitated. As the last of the visible ice melted the cryobag was removed from the water bath and post-thaw analysis began immediately. The duration of the thawing episode was recorded.

For water-free thawing, a controlled-rate thawing station (VIA Thaw, Cytiva, Cambridge, UK) was programmed with the cryobag volume and warmed to 34˚C before the thawing cycle was started. This system uses adaptable metal plates heated to a set temperature (34˚C) to warm a cryobag from both sides. Upon removal from the dry shipper the cryobag was immediately placed into the machine and thawing initiated immediately. When completion of thawing was indicated, the cryobag was removed and post-thaw analysis started. The duration of the thawing episode was recorded.

## Post-thaw analysis of cryopreserved leukapheresis samples

**Trypan blue staining.** Cell samples were diluted in trypan blue solution (0.4% trypan blue in 0.9% saline solution, Sigma, Gillingham, UK #T8154), gently agitated and left to stand for 1 minute. Thereafter, sample-blind live/dead cell counts were carried out using a haemocytometer with a minimum of at least 100 nucleated cells counted per sample. Where necessary the cell suspensions were further diluted in Hanks Balanced Salt Solution (HBSS, HyClone, Cytiva, Cramlington, UK #SH30031.03).

Cells that excluded the trypan blue dye were accepted as having an intact, outer membrane and defined as viable. Those cells with the intracellular volume stained blue were accepted as having a compromised membrane and were defined as non-viable. Trypan blue viability was calculated as the percentage of the cell population with an intact cell membrane.

**Total nucleated cell count.** Total nucleated cells in a 1ml sample of thawed cell preparation were counted immediately post-thaw using an automated H500 cell counter (Yumizen H1500, Horiba, Kyoto, Japan).

**Colony forming units.** A 0.2ml aliquot of cell suspension was placed into 6ml of Methocult gel (Stemcell Technologies, Vancouver, Canada,) and vortexed for 1 minute to allow for full mixing. Samples were allowed to stand for 5 minutes before 1.1ml was placed into each of 4 wells of a 6-well plate. The plate was incubated at 37˚C in 5% $CO_2$ for 14 days in a humidified incubator, at which point a colony count was carried out for each well. A colony was defined as a grouping of approximately 50 or more cells.

**Flow cytometry.** CD45+, CD34+, and CD34+/7-AAD positive cells were counted by flow cytometry as a measure of viable cells [22]. Following a total nucleated-cell count, outlined above, samples were diluted to 1-2x$10^7$ cells ml$^{-1}$ and incubated with CD45 FITC/CD34 PE antibody (BD Bioscience, Wokingham, Berkshire, UK, #341071) and 7-AAD viability dye (BD Bioscience, #559925) in BD Trucount tubes (BD Bioscience, #555899) for 15 minutes at room temperature in the dark. Red cell lysis was performed using Pharmlyse (BD Bioscience, #555899) for 15 minutes. Flow cytometric analysis was performed on a minimum of 100,000 total CD45 positive events using a BD FACSCanto II flow cytometer (BD Bioscience). Absolute live/dead CD45/ CD34 positive cell numbers were determined using a single platform technique with Trucount beads and an ISHAGE Boolean gating strategy selecting CD45/CD34 positive cells with 7-AAD live/dead cell determination [23].

## Statistical analyses

The R software (versions 3.4.2 and 4.0.2) and R Commander 2.4–1 package were used for statistical analyses and displaying the data [24,25].

To compare both thawing methods on the measured CD45+ and CD34+ cell post-thaw recoveries, Bland-Altman analyses were performed, after ensuring the differences between

both thawing methods for each cellular parameter were normally distributed (Shapiro-Wilk normality test, $p$-values = 0.3 and 0.2, respectively).

Linear regressions were used to test for a relationship between post-thaw cellular parameters and either cryogenic storage time or patient age compared with and Pearson correlation coefficient. To investigate the influence of gender and thawing method on post-thaw cellular parameters, box and whisker plots were drawn. Means were compared with the parametric T-test if the samples followed a normal distribution and had homogeneous variances ($p$-values > 0.05) or with the non-parametric Wilcoxon rank sum test if not ($p$-values < 0.05).

## Results

### Thawing method and post-thaw viability of cryopreserved leukapheresis samples

Analysis of post-thaw cell outcome showed no significant differences ($p$-values > 0.05) between water-free and wet thawing (Figs 1 and 2), whether measured as cell viability by the trypan blue dye exclusion method (Fig 1A), total viable CD34+ cells (Fig 1B) or as colony-forming units (Fig 1C). The comparison of water-free and wet thawing methods on CD45+ and CD34+ cell post-thaw viabilities is shown in Fig 2. For CD45+ post-thaw cell viability, the water-free thawing method gave on average lower results than the wet thawing method with a correlation coefficient of 0.961. On the contrary, for CD34+ post-thaw cell viability, the water-free thawing method gave on average higher results than the wet thawing method with a correlation coefficient of 0.834.

Large patient-to-patient variations were observed as is not uncommon in myeloma patients [16,17] e.g. trypan blue dye exclusion ranged from 38% to 100% and CD34+ viability from 8% to 87%. Average trypan blue viability was 69.4 ± 19.2%, which was significantly higher than the CD34+ viability at 49.2 ± 20.4%, ($p$-value < 0.05). Thawing time was measured as 405 ± 101s and 337 ± 121s for the water-free and wet thawing respectively ($p$-value < 0.05).

### Storage time and post-thaw survival

Examination of a potential effect of cryogenic storage duration on cell viability, was determined both as trypan blue dye exclusion and by flow cytometry for CD34+ cells. Considering the combined data for water-free and wet thawed samples (Fig 3), there was no indication of a positive

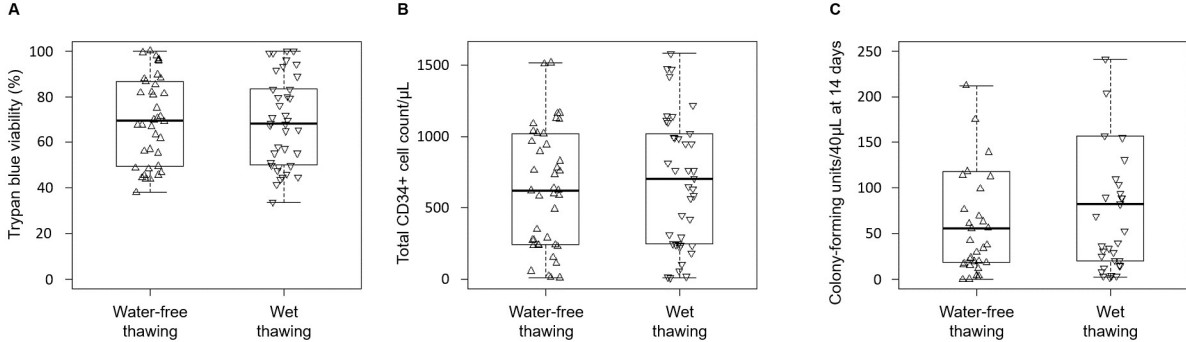

**Fig 1. The comparative effect of water-free and wet thawing on the post-thaw outcome of haematopoietic cells from cryopreserved leukapheresis samples.** (A) Cell viability immediately post-thaw determined as trypan blue exclusion (B) Total CD34+ cell counts (through flow analysis). (C) Colony forming units counted after 14 days post-thaw incubation. The $p$-values obtained from comparing means between thawing methods are indicated.

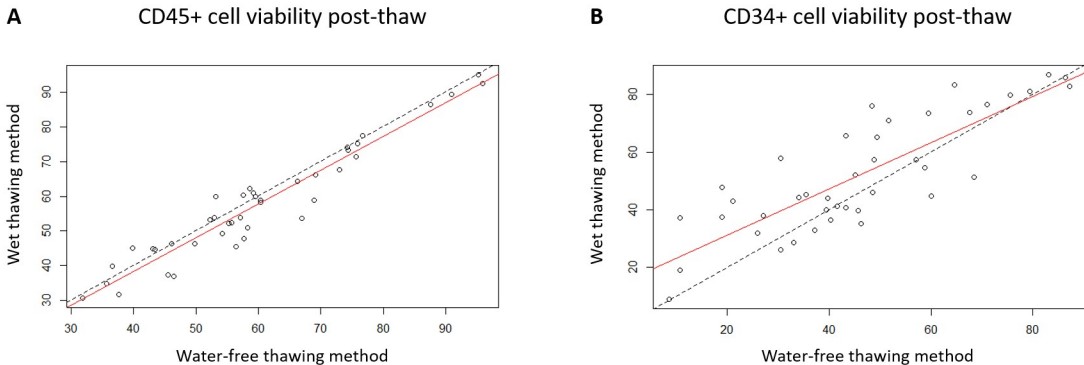

**Fig 2. Bland-Altman analysis of the two different thawing methods.** Plots of post-thaw viability, comparing wet and water free thawing methods with line of equality for (A) CD45+ and (B) CD34+ post-thaw viabilities, determined through flow analysis.

or negative linear relationship between viability and frozen storage time ($p$-values = 0.685 and 0.524 for trypan blue and flow cytometry assessed viabilities, respectively), with a poor representation of the data by the linear model (the adjusted square of the Pearson correlation coefficient, adj. R-squared < 0). Similarly, when the water-free and wet thawing data were considered separately no significant relationship was found: for water-free thawing the $p$-values were 0.944 and 0.631 for trypan blue and flow cytometry assessed viabilities, respectively, and for wet thawing, the comparable $p$-values were 0.532 and 0.677, with negative adj. R-squared values.

## Patient age and post-thaw survival

Patient age at the time of initial collection of blood for leukapheresis, ranging from 39 to 70 years old, had no positive or negative relationship on the immediate post-thaw trypan blue viability ($p$-value = 0.899) when considering the combined data from water-free and wet thawing (Fig 4A). The relationship determined by flow cytometry for post-thaw CD34+ viability (Fig 4B),

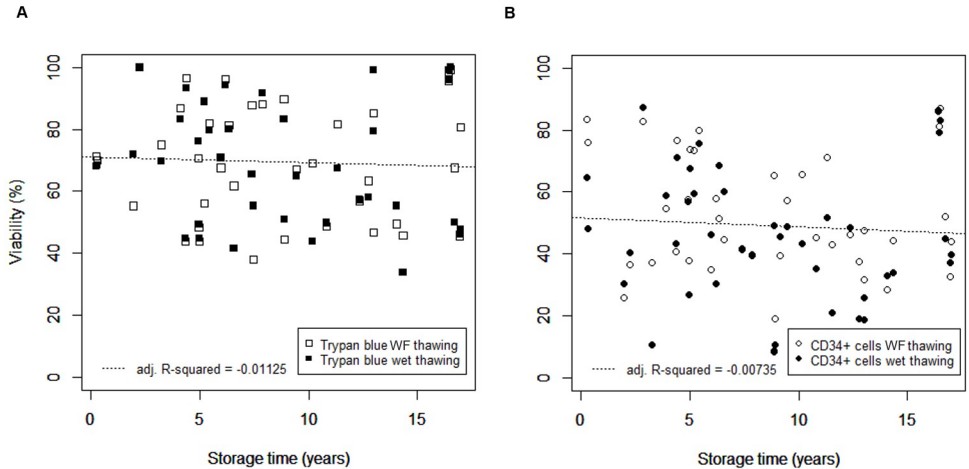

**Fig 3. The comparative effect of cryogenic storage time of cryopreserved leukapheresis samples on the immediate post-thaw cell viability.** (A) Immediate post-thaw cell viability determined as trypan blue exclusion or (B) by flow cytometry for CD34+ cells. Storage periods ranged between 3 months and 17 years, and results for water-free (WF) and wet thawing are presented as open and filled symbols, respectively. The linear model applied to the data as a function of cryogenic storage time is indicated by a dashed line. The adjusted square of the Pearson correlation coefficient (adj. R-squared) indicating the extent of variability in the dataset explained by the linear model, is provided.

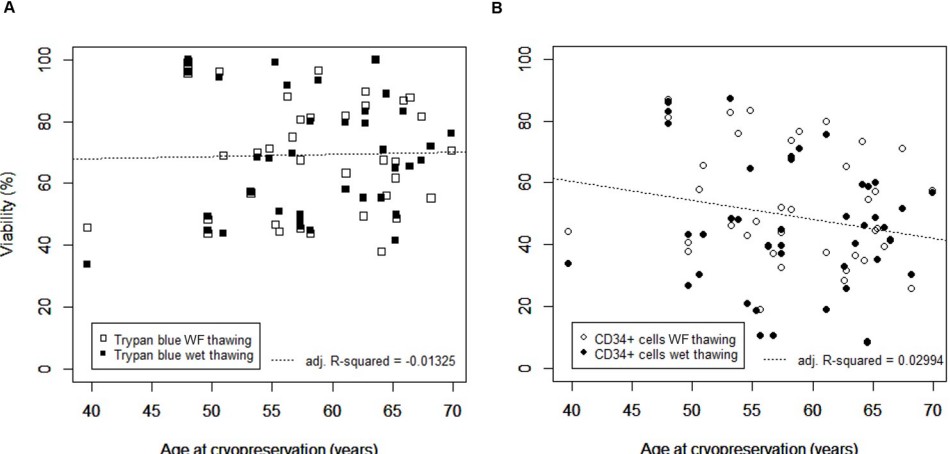

**Fig 4. The effect of patient age at initial collection of leukapheresis samples before cryopreservation on immediate post-thaw cell viability.** (A) Immediate post-thaw cell viability determined as trypan blue exclusion or (B) by flow cytometry for CD34+ cells. Results for water-free and wet thawing are presented as open and filled symbols, respectively. Linear models applied to the combined water-free (WF) and wet thawing datasets as a function of patient age are shown as dashed lines, and the adjusted square of their Pearson correlation coefficient (adj. R-squared) indicating the percentage of variability in the dataset explained by the linear model, is indicated.

however, suggested a negative trend but this was not significant ($p$-value = 0.065). The linear model explained only approximately 3% of the variability in the data (adj. R-squared = 0.0029). When considering the data for water-free thawing, similar observations were made with $p$-values of 0.970 and 0.065 for trypan blue and flow cytometry assessed viabilities, respectively. Similarly, wet thawing gave no significant result for the trypan blue and flow cytometry assessments ($p$-values = 0.820 and 0.429 respectively, with negative adj. R-squared values; data not shown).

## Patient gender and post-thaw survival

Post-thaw viability determined as trypan blue dye exclusion and by flow cytometry for CD45 + and CD34+ cells are presented in Fig 5 together with trypan blue exclusion for all cells. The mean viabilities are not statistically significant ($p$-value > 0.05) between male (n = 23) and female (n = 15 trypan blue; n = 18 CD34+ and CD45+) patients, despite the apparent, higher recovery for female patients.

## Discussion

This study has shown that dry thawing is applicable to cellular materials such as leukapheresis samples, resulting in comparable, correlated, post-thaw outcomes to those produced by an experienced operative using wet thawing (Figs 1 and 2). This was despite the longer time taken to complete thawing in the water-free thawing system when compared to the conventional water bath technique (means of 405 vs. 337 seconds respectively, $p$-value < 0.05). This result may appear unexpected as, across the broader field of cryopreservation, rapid thawing (at least as fast as can be achieved in a 37°C water bath) is considered essential for good post-thaw recovery for a very wide range of cell types [26–29]. However, recent studies have shown that rapid thawing at this level is not required for somatic mammalian cells as long as the earlier cooling stage is appropriately controlled (as is the case with apheresis samples). Damage on warming these samples is commonly caused by the expansion of incomplete ice crystals as more energy becomes available for water mobility. While the

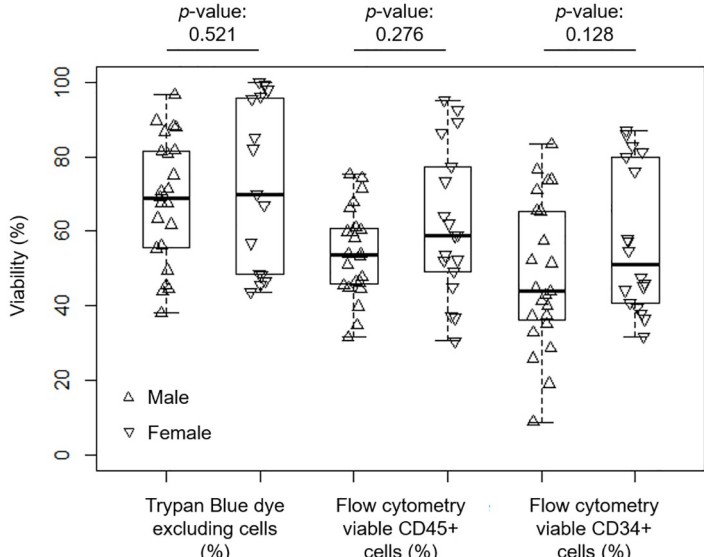

**Fig 5. The comparative effect of patient gender on the immediate post-thaw viability of cryopreserved leukapheresis samples.** Cell viability was determined as trypan blue exclusion or by flow cytometry for CD45+ and CD34+ cells. The *p*-values obtained from comparing means between patient gender are indicated.

sample is still cold enough to support ice, water molecules will more readily migrate to the surfaces of incomplete crystals to form more ice. Slow cooling, below about $10\,^{\circ}C$ min$^{-1}$ in DMSO-based cryoprotectants, allows complete ice formation on cooling and so largely removes the requirement for rapid warming [6,11–13].

The rationale underlying rapid thawing can be illustrated by considering cryopreserved sperm cells. In this instance rapid cooling is employed, together with a glycerol-based cryoprotectant. A consequence of this rapid cooling is that water loss from the cells is limited by diffusion and so the amount of extracellular ice is less than would be expected if an equilibrium had been reached. This means that ice can crystallise during thawing, as described above, causing a potentially lethal osmotic shock for the cells [30]. Rapid thawing limits the extent of this ice formation during thawing, so reducing any damaging, osmotic stresses. However, cryopreserved apheresis samples represent a system with significantly different properties. The relatively slow cooling rate and low viscosity, DMSO-based cryoprotectant that are employed allow more time for diffusion and allow the maximum amount of ice to form during controlled cooling. Consequently, ice crystallisation during thawing will be limited and so thawing can occur rapidly or slowly with a minimal risk of osmotic stress. However, it is important that samples are either used immediately or the DMSO washed out immediately after thawing. DMSO is toxic to cells at higher temperatures, and so thawed cells left in an aqueous DMSO solution will be adversely affected [31]. Establishing that water-free thawing is as effective as wet thawing is critical in enacting GMP processes in the manufacture of cell therapies, as water-free thawing allows for user-independent, traceable, and more accurately recordable thawing profiles, both for the final cell therapy but also for early stages in the manufacture—e.g. thawing of an initial apheresis sample which may be shipped cryopreserved to a manufacturing site as a starting material for the treatment.

The data presented in Fig 3 indicates that extending frozen storage from 3 months to 17 years has no significant effect on the post-thaw outcome of the samples. To ensure the safe, long-term storage of apheresis samples it is critical that the samples are held, continuously,

below the glass transition temperature of the cryoprotectant solution, some -120˚C for DMSO-based solutions [6,32]. Studies with other biological systems have shown this to be effective, and necessary, for decades [4,33–37]. This is achieved as cellular, chemical and biological processes in the sample effectively stop below this temperature [32]. Allowing the storage conditions to rise above the glass transition temperature, even briefly, introduces the risk of resumed diffusion, threatening the stability of the samples [38–40]. It should also be noted that the trypan blue assay gave a higher level of viability than flow cytometry in this instance, as was the case when assessing the effect of gender on post-that performance (Fig 5). This assay is quick and inexpensive, and its use is commonplace, but this potential overestimate of viability should be held in mind when calculating potential cell numbers that can be transplanted.

Unwanted storage temperature fluctuations may occur where many samples are stored together and retrieving one requires moving others, inadvertently exposing them to a temperature rise. If this excursion goes above the glass transition temperature for any particular sample, then a risk to stability arises. The few reports of decline in post-thaw outcome after storage in liquid nitrogen vapour (below -120˚C) are likely due to such unintentional warming. Multiple temperature cycling between the vapour phase of liquid nitrogen and up to -120˚C has been shown to be minimally damaging for PBMC cells [41], but similar studies for apheresis samples are lacking.

There was no significant negative impact of patient age on post-thaw assessment (Fig 4) up to 70 years, the maximum within the study, which agrees with findings for healthy donors [42]. This would appear to support continuing with the protocol, without modification, for the greater proportion of the patient population. However, a possible trend was observed for CD34+ cells where increased patient age seemed to negatively impact post-thaw viability (Fig 4B), and it may be possible that this trend became significant if the dataset included more patients aged 70 and above. Myeloma cases are more common in older patients with an average age of 67 and reduced engraftment has been observed in patients of 70+ years, particularly with respect to CD34+ cells [42,43]. This may become more significant concern as survival rates improve and the upper age limit of the patient population increases. The underlying reasons for this need to be determined and their significance for research into possible changes in the cryopreservation protocol considered.

Patient gender also had no significant impact on the post-thaw outcome of cells. Few studies have been reported looking specifically at the differences between male and female cryopreservation outcome. From this study of peripheral blood mononuclear cells, any improved outcome for female-derived samples did not stand up to statistical scrutiny.

## Acknowledgments

We are grateful to the Cambridge Cellular Therapy Laboratory staff for their help and support throughout this study.

## Author Contributions

**Conceptualization:** Peter Kilbride, Julie Meneghel, G. John Morris, Kevin Jestice.

**Data curation:** Peter Kilbride, Julie Meneghel, Giovanna Creasey, Fatemeh Masoudzadeh, Tina Drew, Hannah Creasey, Kevin Jestice.

**Formal analysis:** Peter Kilbride, Julie Meneghel.

**Investigation:** Peter Kilbride, Julie Meneghel, Giovanna Creasey, Fatemeh Masoudzadeh, Tina Drew, Hannah Creasey, Kevin Jestice.

**Methodology:** Peter Kilbride, Julie Meneghel, Hannah Creasey, Kevin Jestice.

**Project administration:** Peter Kilbride, Julie Meneghel, Tina Drew, Hannah Creasey, G. John Morris, Kevin Jestice.

**Resources:** G. John Morris, Kevin Jestice.

**Supervision:** Peter Kilbride, Julie Meneghel, David Bloxham, G. John Morris.

**Writing – original draft:** Peter Kilbride, Julie Meneghel.

**Writing – review & editing:** Peter Kilbride, Julie Meneghel, Hannah Creasey, David Bloxham, Kevin Jestice.

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
