## [Decision Letter · Decision Letter 0]

11 Aug 2020

PONE-D-20-17327

Automated dry thawing of cryopreserved haematopoietic cells is not adversely influenced by cryostorage time, patient age or gender

PLOS ONE

Dear Dr. Kilbride,

Thank you for submitting your manuscript to PLOS ONE. After careful consideration, we feel that it has merit but does not fully meet PLOS ONE’s publication criteria as it currently stands. Therefore, we invite you to submit a revised version of the manuscript that addresses the points raised during the review process.

There is obviously a large differences of opinion between the two reviews.  Please try to address the critical issues raised by the one reviewer. 

We look forward to receiving your revised manuscript.

Kind regards,

Jeffrey Chalmers, Ph.D.

Academic Editor

PLOS ONE

Journal Requirements:

2. Thank you for including your competing interests statement; "I have read the journal's policy and the authors of this manuscript have the following competing interests: Authors PK, JM, GC, and GJM are employees of Cytiva."

We note that one or more of the authors are employed by a commercial company: Cytiva, Danaher Corporation, Cambridge, UK

Reviewers' comments:

Reviewer's Responses to Questions

**Comments to the Author**

1. Is the manuscript technically sound, and do the data support the conclusions?

Reviewer #1: No

Reviewer #2: Yes

2. Has the statistical analysis been performed appropriately and rigorously? 

Reviewer #1: No

Reviewer #2: Yes

3. Have the authors made all data underlying the findings in their manuscript fully available?

Reviewer #1: Yes

Reviewer #2: Yes

4. Is the manuscript presented in an intelligible fashion and written in standard English?

Reviewer #1: Yes

Reviewer #2: Yes

5. Review Comments to the Author

Reviewer #1: In this study cryopreserved cell viability after dry and wet thawing is compared. No differences between both methods are observed in a comparison of obtained viabilities by t-test/Wilcoxon SR test. Furthermore, no correlation between viabilities and storage time, age, or sex is observed. The authors interpret these observations as being supportive for the use of dry thawing.

General:

The choice of statistical analysis in this study is not convincing. Instead of a t-test or Wilcoxon SR test, other methods (e.g., linear regression, Bland-Altman analysis) should have been applied. CD34+ viability ranged between ~10% and ~90% for both methods with a quite homogenous distribution and a possible range of 0-100%. It is obvious that P would not be <.05. The current way of presentation does not allow pairwise comparison of dry and wet thaw results on an individual basis. The attentive reader, however, can see in Fig. 2,3 that viability differs by up to 20-30 percent points in some cases (e.g., the ~64-year-old individual whose cells were stored for ~3 years). This suggests poor agreement between both methods of thawing, which in turn would impair assessment of the subgroup analyses.

Abstract:

The abstract should adhere to the conventional objectives-methods-results-conclusions structure in compliance with the journal author guidelines. Currently, it does not include information on background/objectives.

Introduction:

- I understand that all analyzed samples were cryopreserved PBMCs from stimulated donors (although this is never explicitly stated). Therefore, the introduction should be shortened, and it should focus on these products. While being the hot topic of the time, CAR-T cells have not much to do with this study.

- In the last paragraph the authors state that the studied samples were from “donors treated for myeloma and now in remission”, and that 2x10^6 viable cells/kg BW already had been used for therapy prior to the study. In case of the products with ~10% CD34+ viability there must have been a unplausibly high number of cryobags available to achieve this therapeutic dose. I am assuming that the viability of the cells used for infusion was as low as the viability observed in this study, as the authors state that cryogenic storage time does not affect post-thaw viability.

Materials and Methods:

- It should be explicitly stated what products are studied, e.g., PBMCs from autologous donors after stimulation (GCSF only? GCSF plus plerixafor?). It should be explicitly stated, if the cryobags used for comparison were identical pairs (Same total volume? Same content?)

- The dry thawing device should be described in more detail, as the reader might not be familiar with it.

Results:

- Viability is unusually low in a quite high number of products. Pre-freeze viability usually is near 100%. In the literature recovery rates of viable CD34+ cells are usually described to lay between 80% and 90%. In real life recovery rates might be as low as 50-60% in some cases. But viabilities of 10-20% implicate recovery rates of <50%, and it is not clear to me, how one could successfully collect enough CD34+ cells for therapeutic use with such low viability.

- In the last paragraph the authors state that trypan blue viability was higher than CD34+ and CD45+ viability, but no statistical comparison is provided.

Discussion:

- First paragraph: The absence of a significant difference in this case does not in turn implicate that the compared methods are equivalent (see general comment).

- Second paragraph discusses importance of thaw time without much concrete reference to the performed analyses and observed results.

- Third paragraph: See above (second comment regarding Results section).

- Fourth paragraph: Again, too much general information.

- Sixth paragraph: “Red blood cell transplant effectiveness” and ref. 43 refer to common allogenic RBC transfusions (without cryopreservation). These cannot be compared with autologous PBMC infusions.

Author contributions:

- The comment used for internal communication between the authors should be removed from the word document before submission.

Reviewer #2: This study compared two methods for thawing hematopoietic stem cell grafts cryopreserved using DMSO, thawing using the traditional water bath and thawing using an automated water-free thawing device. The study also evaluated the effect of storage duration, patient age and patient gender on the post-thaw recovery of leukocytes and CD34+ cells. While all of the data is important to laboratories involved with processing hematopoietic stem cells, the most relevant data involves the finding that automated water-free thawing device yielded similar results as a water bath.

These finding are of particular importance to laboratories thawing genetically engineered T-cells such as chimeric antigen receptor (CAR) T-cells. The processing of CAR T-cells is, in general, expected to meet strict Good Manufacturing Practice requirements. Thawing cells using the automated water-free thawing device prevents variations in the thawing process due to technique differences among lab staff which allows for a more consistent thawing process. In addition, maintaining a water bath in a processing laboratory presents a risk for microbial contamination of the laboratory and the product. The study did not directly evaluate thawing of genetically engineered T cells, but the results of thawing with the automated water-free thaw device would likely be similar. While the manuscript mentions the advantages of the automated water-free thaw device in the introduction section, it may be worthwhile to include some mention of these advantages in the discussion section.

6. PLOS authors have the option to publish the peer review history of their article (what does this mean?). If published, this will include your full peer review and any attached files.

Reviewer #1: No

Reviewer #2: No

---

## [Author Response · Author response to Decision Letter 0]

4 Sep 2020

Dear Editor,

Please note that these have also been uploaded as a word document.

Thank you for giving us the opportunity to revise our work for PLOS. We’d also like to thank the reviewers for their helpful comments, and we hope that you find the revised text suitable for publication. In the below document, we first answer the journal requests regarding competing interests and the financial statement, before going on to answer the reviewer comments. We have kept the reviewer comments in black, with our responses in blue. 

Best Wishes,

Peter and the authors,

1. We have prepared a Financial Statement as requested:

The funder Cytiva provided support in the form of salaries only for authors PK, GC, JM, and GJM, but did not have any additional role in the study design, data collection and analysis, decision to publish, or preparation of the manuscript. The specific roles of these authors are articulated in the ‘author contributions’ section.

And an updated Competing Interests Statement as requested:

Authors PK, JM, GC, and GJM are employees of Cytiva, which provided salaries for these authors. No consultancy, patents, products in development, or marketed products were derived from this study. This does not alter our adherence to PLOS ONE policies on sharing data and materials. 

Responses to reviewers:

Reviewer #1: In this study cryopreserved cell viability after dry and wet thawing is compared. No differences between both methods are observed in a comparison of obtained viabilities by t-test/Wilcoxon SR test. Furthermore, no correlation between viabilities and storage time, age, or sex is observed. The authors interpret these observations as being supportive for the use of dry thawing.

General:

The choice of statistical analysis in this study is not convincing. Instead of a t-test or Wilcoxon SR test, other methods (e.g., linear regression, Bland-Altman analysis) should have been applied. CD34+ viability ranged between ~10% and ~90% for both methods with a quite homogenous distribution and a possible range of 0-100%. It is obvious that P would not be <.05. The current way of presentation does not allow pairwise comparison of dry and wet thaw results on an individual basis. The attentive reader, however, can see in Fig. 2,3 that viability differs by up to 20-30 percent points in some cases (e.g., the ~64-year-old individual whose cells were stored for ~3 years). This suggests poor agreement between both methods of thawing, which in turn would impair assessment of the subgroup analyses.

We agree that for these age and storage time tests a linear regression is appropriate for population-wide tests i.e. for when we plot say storage time or patient age against post-thaw viability. We have in fact done this for these datasets shown in figures 2 and 3 (now numbered figures 3 and 4 in the revised manuscript), using a Pearson correlation coefficient to a linear regression model. We have changed the text to be more explicit about the method used here (p. 8, l. 216-217 in the revised manuscript with track changes). 

We agree that Bland-Altman analyses, as suggested by the reviewer, on CD45+ and CD34+ post-thaw cell viabilities relative to wet and dry thawing is appropriate as the differences between both methods for each cellular parameter were normally distributed (test performed: Shapiro-Wilk normality test, p-values = 0.3 and 0.2, respectively). We have carried out these analyses using Bland-Altman as suggested by the reviewer, and the plot of both thawing methods with line of equality for CD45+ and CD34+ post-thaw cell viabilities are shown in the updated Figure 1. 

Overall, agreement between both thawing methods is considered substantial for CD45+ and moderate for CD34+ (Altman, 1991; McBride, 2005), which was confirmed by concordance correlation coefficients (Lin et al. 1989), with values of 0.95 and 0.80, respectively. We have updated the methods (p. 8, l. 210-214), figure numbering (throughout the results and discussion sections), and results section accordingly (p. 9, l. 234-238). 

Abstract:

The abstract should adhere to the conventional objectives-methods-results-conclusions structure in compliance with the journal author guidelines. Currently, it does not include information on background/objectives.

We agree and have added some new text to the start of the abstract to better put the work and aims in context (p. 2, l. 28-34).

Introduction:

- I understand that all analyzed samples were cryopreserved PBMCs from stimulated donors (although this is never explicitly stated). Therefore, the introduction should be shortened, and it should focus on these products. While being the hot topic of the time, CAR-T cells have not much to do with this study.

We’ve updated the methods section to make the mobilization of the cells, and this general process, more detailed and reproducible (also based on another comment below; p. 5, l. 125-128). We have removed some extra text in the introduction where possible (p. 3, l. 72-76). We have also removed some references to T cells too (p. 3, l. 61-65) to streamline the introduction, although we think it is relevant to keep some mention of the CAR-T area. This is because CAR-T is perhaps the fastest growing area of research in apheresis samples, and so we think that there is relevance and usefulness from this work to researchers in that area. Specifically, in CAR-T processing the initial step usually involves taking an apheresis sample from a patient, cryopreserving and shipping it, before thawing at a manufacturing site, a process for which this work is directly relevant (except perhaps the extended storage times part). While this paper focuses on apheresis samples which were taken for use in myeloma samples, this doesn’t preclude the relevance for the data for apheresis samples used in other areas.

- In the last paragraph the authors state that the studied samples were from “donors treated for myeloma and now in remission”, and that 2x10^6 viable cells/kg BW already had been used for therapy prior to the study. In case of the products with ~10% CD34+ viability there must have been a unplausibly high number of cryobags available to achieve this therapeutic dose. I am assuming that the viability of the cells used for infusion was as low as the viability observed in this study, as the authors state that cryogenic storage time does not affect post-thaw viability.

We have rephrased this statement in the text as it was unacceptably ambiguous (p. 5, l. 112-113). What we intended to state in this paragraph was that the myeloma samples had been taken from patients who in the past had been successfully treated and so were in remission at point of apheresis. The 2x10^6 viable cells/kg is actually the pre-freeze value target (this was reworded in p. 5, l. 115-117). In practice we thaw and transfuse cells immediately, as the cells start to die due to the DMSO toxicity as soon as they are warmed so we cannot wait for flow work. In the case of very low viabilities (trypan blue as flow takes too long), we advise the clinical who can decide the best course of action (such as give a larger number of cryobags for example). The work shows that cryogenic storage duration does not significantly affect post-thaw viability, however the freeze-thaw process itself does, to an extent that is minimal to considerable (with for instance CD34+ cell viability ranging from 8% to 87%), due to patient sample variability at apheresis.

Materials and Methods:

- It should be explicitly stated what products are studied, e.g., PBMCs from autologous donors after stimulation (GCSF only? GCSF plus plerixafor?). It should be explicitly stated, if the cryobags used for comparison were identical pairs (Same total volume? Same content?)

- The dry thawing device should be described in more detail, as the reader might not be familiar with it.

We have edited the methods section to give more detail about the mobilization process of the patients (p. 5, l. 125-128), as well as making clear that all bags were identical (p. 6, l. 143-144). We have added another reference about dry thawing devices and explained further how they work in the introduction (p.4, l. 96-98 and 103-104) and in the methods (p.6, l. 162-163). One thing we want to avoid in this paper is making it too specifically focused on a single thawing device – the main outcome of this part of the work is that water-free thawing has the same outcome as a water bath, even though the thaw takes a little longer (which would be true for any dry thawing system as thermal transfer relies on conduction rather than on convection) – we believe these results aren’t limited to the system tested here so don’t want to limit the readers’ interest. The details of the system are mentioned in the text, so a reader should find out more detailed specifics if required. 

Results:

- Viability is unusually low in a quite high number of products. Pre-freeze viability usually is near 100%. In the literature recovery rates of viable CD34+ cells are usually described to lay between 80% and 90%. In real life recovery rates might be as low as 50-60% in some cases. But viabilities of 10-20% implicate recovery rates of <50%, and it is not clear to me, how one could successfully collect enough CD34+ cells for therapeutic use with such low viability.

These are taken from a patient in the expectation of having enough cells, with a target of 2X10^6 cells/kg body weight minimum taken from each patient. However with some patients, either because they’re very ill, have a poor response to mobilization, or have a compromised immune system etc. have cells which freeze poorly (the exact mechanism by which one patient has ‘better’ cells in terms of freezing related to another is poorly understood), but we’ve highlighted this variation in results (p. 9, l. 239-240) and in the first section of the discussion (p. 13, l. 337-341). We didn’t want to remove these very poor results as this might artificially boost the suggested numbers, however we agree that these patients would have a low chance of engraftment – in practice we usually advise the clinician in the cases of very low viability, however infusing lower than 2x10^6 cell/kg doesn’t necessarily mean there won’t be engraftment. 

- In the last paragraph the authors state that trypan blue viability was higher than CD34+ and CD45+ viability, but no statistical comparison is provided.

We have moved this text to the first section of the results where we think this sits better, and also added in the statistical comparison (p. 9, l. 239-240).

Discussion:

- First paragraph: The absence of a significant difference in this case does not in turn implicate that the compared methods are equivalent (see general comment).

See response to first comment.

- Second paragraph discusses importance of thaw time without much concrete reference to the performed analyses and observed results.

We have now added the thawing data to the results section to make the discussion better placed (p. 9, l. 240-241).

- Third paragraph: See above (second comment regarding Results section).

Answer same as previous.

- Fourth paragraph: Again, too much general information.

We have removed the general information to shorten the paragraph (p. 13, l. 330-335).

- Sixth paragraph: “Red blood cell transplant effectiveness” and ref. 43 refer to common allogenic RBC transfusions (without cryopreservation). These cannot be compared with autologous PBMC infusions.

This reference has been removed, and the final paragraph of this section re-written (p. 14, l. 363-371).

Author contributions:

- The comment used for internal communication between the authors should be removed from the word document before submission.

This has been removed.

Reviewer #2: This study compared two methods for thawing hematopoietic stem cell grafts cryopreserved using DMSO, thawing using the traditional water bath and thawing using an automated water-free thawing device. The study also evaluated the effect of storage duration, patient age and patient gender on the post-thaw recovery of leukocytes and CD34+ cells. While all of the data is important to laboratories involved with processing hematopoietic stem cells, the most relevant data involves the finding that automated water-free thawing device yielded similar results as a water bath.

These finding are of particular importance to laboratories thawing genetically engineered T-cells such as chimeric antigen receptor (CAR) T-cells. The processing of CAR T-cells is, in general, expected to meet strict Good Manufacturing Practice requirements. Thawing cells using the automated water-free thawing device prevents variations in the thawing process due to technique differences among lab staff which allows for a more consistent thawing process. In addition, maintaining a water bath in a processing laboratory presents a risk for microbial contamination of the laboratory and the product. The study did not directly evaluate thawing of genetically engineered T cells, but the results of thawing with the automated water-free thaw device would likely be similar. While the manuscript mentions the advantages of the automated water-free thaw device in the introduction section, it may be worthwhile to include some mention of these advantages in the discussion section.

We have added some more detail in the discussion (p. 12, l. 313-318). To avoid this being a repeat of the introduction, we have mentioned some of the GMP practical considerations related to dry thawing, which is likely useful to many clinical readers.

---

## [Decision Letter · Decision Letter 1]

24 Sep 2020

Automated dry thawing of cryopreserved haematopoietic cells is not adversely influenced by cryostorage time, patient age or gender

PONE-D-20-17327R1

Dear Dr. Kilbride,

We’re pleased to inform you that your manuscript has been judged scientifically suitable for publication and will be formally accepted for publication once it meets all outstanding technical requirements.

Kind regards,

Jeffrey Chalmers, Ph.D.

Academic Editor

PLOS ONE

Additional Editor Comments (optional):

Reviewers' comments:

Reviewer's Responses to Questions

**Comments to the Author**

1. If the authors have adequately addressed your comments raised in a previous round of review and you feel that this manuscript is now acceptable for publication, you may indicate that here to bypass the “Comments to the Author” section, enter your conflict of interest statement in the “Confidential to Editor” section, and submit your "Accept" recommendation.

Reviewer #2: (No Response)

2. Is the manuscript technically sound, and do the data support the conclusions?

Reviewer #2: Yes

3. Has the statistical analysis been performed appropriately and rigorously? 

Reviewer #2: I Don't Know

4. Have the authors made all data underlying the findings in their manuscript fully available?

Reviewer #2: Yes

5. Is the manuscript presented in an intelligible fashion and written in standard English?

Reviewer #2: Yes

6. Review Comments to the Author

Reviewer #2: The authors have addressed all of the comments made by this reviewer. I have no further concerns about the manuscript.

7. PLOS authors have the option to publish the peer review history of their article (what does this mean?). If published, this will include your full peer review and any attached files.

Reviewer #2: No

---

## [Editor Report · Acceptance letter]

15 Oct 2020

PONE-D-20-17327R1 

Automated dry thawing of cryopreserved haematopoietic cells is not adversely influenced by cryostorage time, patient age or gender 

Dear Dr. Kilbride:

I'm pleased to inform you that your manuscript has been deemed suitable for publication in PLOS ONE. Congratulations! Your manuscript is now with our production department. 

Kind regards, 

on behalf of

Dr. Jeffrey Chalmers 

Academic Editor

PLOS ONE